# Differentially Expressed Cell Cycle Genes and STAT1/3-Driven Multiple Cancer Entanglement in Psoriasis, Coupled with Other Comorbidities

**DOI:** 10.3390/cells11233867

**Published:** 2022-11-30

**Authors:** Subhashini Dorai, Daniel Alex Anand

**Affiliations:** The Centre for Molecular Data Science and Systems Biology, Department of Bioinformatics, School of Bio and Chemical Engineering, Sathyabama Institute of Science and Technology, Deemed to be University, Chennai 600119, India

**Keywords:** psoriasis comorbidity, cell cycle proteins, STAT1/3, cancer development, remedies

## Abstract

Psoriasis is a persistent T-cell-supported inflammatory cutaneous disorder, which is defined by a significant expansion of basal cells in the epidermis. Cell cycle and STAT genes that control cell cycle progression and viral infection have been revealed to be comorbid with the development of certain cancers and other disorders, due to their abnormal or scanty expression. The purpose of this study is to evaluate the expression of certain cell cycle and STAT1/3 genes in psoriasis patients and to determine the types of comorbidities associated with these genes. To do so, we opted to adopt the in silico methodology, since it is a quick and easy way to discover any potential comorbidity risks that may exist in psoriasis patients. With the genes collected from early research groups, protein networks were created in this work using the NetworkAnalyst program. The crucial hub genes were identified by setting the degree parameter, and they were then used in gene ontology and pathway assessments. The transcription factors that control the hub genes were detected by exploring TRRUST, and DGIdb was probed for remedies that target transcription factors and hubs. Using the degree filter, the first protein subnetwork produced seven hub genes, including STAT3, CCNB1, STAT1, CCND1, CDC20, HSPA4, and MAD2L1. The hub genes were shown to be implicated in cell cycle pathways by the gene ontology and Reactome annotations. The former four hubs were found in signaling pathways, including prolactin, FoxO, JAK/STAT, and p53, according to the KEGG annotation. Furthermore, they enhanced several malignancies, including pancreatic cancer, Kaposi’s sarcoma, non-small cell lung cancer, and acute myeloid leukemia. Viral infections, including measles, hepatitis C, Epstein–Barr virus, and HTLV-1 and viral carcinogenesis were among the other susceptible diseases. Diabetes and inflammatory bowel disease were conjointly annotated. In total, 129 medicines were discovered in DGIdb to be effective against the transcription factors BRCA1, RELA, TP53, and MYC, as opposed to 10 medications against the hubs, STAT3 and CCND1, in tandem with 8 common medicines. The study suggests that the annotated medications should be tested in suitable psoriatic cell lines and animal models to optimize the drugs used based on the kind, severity, and related comorbidities of psoriasis. Furthermore, a personalized medicine protocol must be designed for each psoriasis patient that displays different comorbidities.

## 1. Introduction

Keratinocytes (KETs) are sensitive to a variety of stimuli that might react to the components of the inborn immune response. During both the initial stages and ongoing stages of psoriasis, KETs perform crucial functions. In the initial stage, strained KETs stimulate plasmacytoid dendritic cells (pDCs) by releasing antimicrobial agents and auto-generated nucleotides. The pDCs then release IFN-α and γ, IL-1β and TNF-α. These molecules trigger and mature myeloid dendritic cells (mDCs). By producing IL-23, TNF and IL-12, the mDC triggers Th17, Th22 and Th1 cells, respectively. Th17 releases IL-17/22, Th22 releases IL-22 and Th1 releases IFN-γ and TNF-α. IL-17/22 begins the hyperproliferation of KETS. In the ongoing stage of psoriasis, KETs act as aggressors of the psoriatic inflammatory response. KETs become extremely proliferative when proinflammatory cytokines have triggered them in a coordinated manner. They can also release a large number of chemokines, such as CXCL1-3/8/9-11 and CCL2/20, which can attract neutrophils, dendritic cells, macrophages and Th17 cells. Antimicrobial substances, such as S100A7-9/12, LL37, and hBD2, are also synthesized to trigger an innate immune response. Furthermore, KETs that produce IL-19/36 induce hyperplasia in the epidermis. In addition, KETs promote tissue remodeling through the induction and multiplication of endothelial cells, along with the accumulation of the extracellular matrix. Endothelial cells and fibroblasts assist in this process. KETs’ irregular development and its hyperproliferation, enlarged and hyperplasic blood vessels and leukocyte influx are all signs of psoriasis pathogenesis. This crosstalk is brought about by the interaction between KETs and Th17 cells [1].

In the epidermis, the excessive proliferation of basal cells distinguishes psoriasis from other skin conditions [2]. Patients with psoriasis have a 120 h turnover period for the epidermal area under the keratin barrier, and a 91 h cell cycle duration for the germ cell chamber. DNA synthesis has been reported to take 10 h during the S phase, and 30 min during mitosis. If the keratin layer’s turnover period is believed to be 2 days, then the epidermis’ turnover time in psoriasis must be 7 days, rather than the formerly mentioned 3–4 days [3].

Psoriasis patients have extensive concerns about their increased risk of developing cancer. Due to the long-term chronic characteristics of the disorder, the usage of immunosuppressants and UV applications, along with the growing frequency of concomitant, visible cancer risk behaviors such as smoking, of which could enhance the possibility of carcinogenesis [4], psoriasis has been linked to an increased likelihood of cancer, especially lymphoma, and individuals with much more extreme symptoms are at a high risk of death from cancer-related causes, as claimed by studies [5,6]. Moreover, these analyses typically did not account for significant mediators and frequently neglected to look at the prevalence of certain cancers or the influence of pathogenicity on cancer development. To better identify the likelihood of malignancies in individuals with psoriasis, additional research is required [4,7]. This would, in any event, assist in the identification of drugs not only to treat the malignancies, but also to therapeutically address psoriasis as a comorbidity.

A recent demonstration has shown that an external source of psoriasis is infection. Various pathogenic agents interact with immune cells to release inflammatory mediators, which can either promote or worsen psoriasis [8]. The infection could be possibly linked to increased sadness and stress in psoriasis patients, although concrete evidence is not yet available [9]. Through the release of cytokines, infections can increase the frequency of psychological symptoms in psoriasis patients. Pro-inflammatory mediators of cytokines, namely IL-6/17, and TNF-α, in psoriasis have been demonstrated to be linked to psychiatric illnesses [10]. The three predominant types of infection that can cause psoriasis are bacterial, viral, and fungal. A definitive remedy for psoriasis is extremely difficult to find because the disease’s pathophysiology has not yet been thoroughly studied [8].

The protein–protein interactions (PPI), which contain a variety of protein–protein linkages and graphically characterize them, can be created using NetworkAnalyst (NA) software [11]. The STRING interactome (SIT) in NA combines PPI data from functional and physical relationships, which originate from computational and experimental predictions for more than two thousand species. The STRING project’s primary point of distinction is that every interaction is given a confidence score (CS). More evidence-based interactions are given high scores. One can modify the CS cut-off to prevent PPIs below the chosen value from being included in the network [12].

TRRUST is a repository of transcription factor (TF)–target regulatory associations discovered by the manual review of abstracts from Medline. With 8015 TF–target interactions, TRRUST is the largest open database of regulatory associations assembled from the literature for humans. Nonetheless, most interactions feature annotations for the mode of activation or repression. The TRRUST source is a helpful baseline for the computative rebuilding of transcription regulatory circuits in humans because gene pairs are strongly weighted between the highest-ranked regulatory connections predicted from high-throughput data sets [13]. TRRUST v2 has several improvements compared to the first version, together with a database with 8444 regulatory associations for 800 human TFs, a considerable improvement from the previous version [14].

Taken together, we have collected 164 differentially expressed genes (DEGs) produced in psoriasis patients from previous experiments to comprehend what type of infections, cancers, or any additional comorbidity emerges alongside psoriasis [15,16,17,18]. NA and TRRUST were implemented in this work to identify the pivotal genes and their corresponding TFs that are responsible for the pathogenesis of psoriasis and its associated comorbidities.

## 2. Materials and Methods

The NA tool (https://www.networkanalyst.ca/; accessed on 15 September 2022) had 164 DEGs loaded, and it used the SIT to build PPIn [11,12]. We utilized the previous (updated on 18 November 2021) and the current version (6 September 2022) of NA. Since all our other research works used the previous version, we continued the analysis from there and finished using the current version [19,20]. The first large subnetwork, SN1, was used. The nodal table was retrieved and the degree measure was used to extract the 10% of DEGs that serve as hub genes from the seeds of SN1. They were then submitted to the batch selection (BAS) option. Once the DEGs were highlighted, the gene ontology and the pathway investigations were conducted by selecting the highlighted nodes and submitting them in the query box of function explorer. The TFs that control the hub genes were discovered using the TRRUST database in the regulation explorer query box. To this end, the standout hub genes and the hallmark TFs were scanned for drugs in the DGIdb (https://www.dgidb.org/; accessed on 20 September 2022) [21].

## 3. Results

The NA platform received the DEGs and constructed PPIn using the SIT [11,12]. Sixteen subnetworks were created, but we focused solely on SN1, as we had already mapped the residual fifteen subnetworks in our prior study [20]. It consisted of 67 seeds, 1136 edges, and 788 nodes (18 November 2021 version). The type of gene expression (standard) was chosen. In Figure 1, the red dots depict the URGs and the green dots depict the DRGs.

The nodal table was downloaded for the SN1 from the NA tool. Based on the degree score, the DEGs were sorted and ranked. In addition, 10% of 67 seeds was fixed as a filtering benchmark to isolate the hub genes [22]. Seven DEGs with the highest degree scores from the highest rank were extracted. STAT3, CCNB1, STAT1, CCND1, CDC20, HSPA4, and MAD2L1 with degree scores of 125, 91, 84, 70, 66, 57, and 51, respectively, were identified as the key hub genes, as displayed in Figure 2. Apart from CCND1, the residual six DEGs were URGs. When compared to other hub genes, STAT3 was the most dominant gene and was ranked the highest in the degree table. After this, the BAS received the hub genes. They were emphasized in the SN1. The results of the GO and pathway studies of the highlighted hub nodes were elucidated using 5% α, the level of significance. Initially, 113 BP, 12 MF, and 26 CC were gathered from the GO analysis. The original pathways obtained were 32, and 60 from KEGG and Reactome. The following number of pathways were obtained when the 5% α was set (Table 1, Table 2, Table 3, Table 4 and Table 5): 31, 1, 5, 17, and 21.

Several DEGs were observed to be associated with drugs. These were STAT3 and CCND1, which matched with 6 and 12 different drugs (Table 6). There were 18 drugs in total that target hub genes in DGIdb [21]. The drugs against the residual DEGs are to be identified in the near future.

The TRRUST database on the NA platform revealed that two-thirds (≥3 out of 7) of the input hub genes controlled by TFs were powerful TFs. Five of them fit this specification (Table 7). This included the transcriptional regulators BRCA1, RELA, TP53, MYC, and STAT3, which regulated five, four, four, three, and three hub genes, respectively. The TFs in the above list did not map to CDC20. The remaining six hub genes, however, were mapped. So, we used TRRUST v2 (https://www.grnpedia.org/trrust/; accessed on 17 September 2022) to identify TFs for CDC20 in humans [14]. PHF8 and YBX1 are the two TFs that regulate CDC20 and have been confirmed with their corresponding PMIDs as 23979597 and 20596676.

Since STAT3 is a hub gene that is also a TF, the drugs for the other TFs were screened in DGIdb because it had already been screened for drugs as a hub gene. BRCA1, RELA, TP53, and MYC were linked to 27, 4, 107, and 21 drugs (Table 8), respectively. Drug interactions were not found for PHF8 and YBX1 in DGIdb. TP53 was found to be associated with the highest number of drugs in contrast to RELA. Altogether, 159 different types of drugs are available to target these TFs.

It was observed that some medications interfere with various TFs. To prevent repetition, even though the same drug has multiple roots and PMIDs, and to correctly compute the overall number of drugs, the intersection of medications versus TFs was evaluated. In addition, 13, 5 and 2 common medicines were expected to work against the TFs (Table 9). When the redundancy was eliminated, there were 137 medications remaining from the 159 drugs listed in Table 8. Furthermore, one, one and six common remedies against hub genes and some TFs were reported (Table 10). On the whole, 8 common drugs, 10 against hub genes and 129 against TFs, were reported, totaling 147 drugs.

## 4. Discussion

Genes that frequently interact with many different genes are referred to as hub genes in gene networks. Hub genes typically occupy an imperative function in cellular processes as an outcome of these interactions [23]. We uncovered seven of these hub genes from the current annotation study and intended to explore how they contributed to the comorbidities reported in psoriasis patients. HSPA4 is a hub gene; however, unexpectedly, the NA did not annotate it in the GO and pathway analyses. With this gene excluded, our discussion is concentrated on the remaining six hub genes and six TFs, as STAT3 is already considered as a hub gene, which was annotated during the inquiry. Based on the previous studies, STAT3, CCNB1, STAT1, CDC20, and MAD2L1 were all upregulated, whilst CCND1 was downregulated [15,16,17,18]. With this in mind, as well as taking into account our result tables, a rational interpretation can be established.

The fundamental task of the cell cycle is to perfectly copy the large quantity of chromosomal DNA, and subsequently split the copies into two daughter cells with the same genetic makeup. The majority of cells need a longer time to duplicate the proportion of organelles and proteins inside of them than they do to replicate and divide. Several gap phases exist in the majority of cell cycles to provide extra time for development. Consequently, the cell cycle of eukaryotes consists of G1, S, G2, and M phases. The interphase is the time between the first three phases. A normal human cell growing in culture might spend 23 of its 24 h in the interphase, leaving 1 h for the M phase [24]. Cyclin genes, such as CCND1, CCNE1, and CCNB1, operate sequentially to control the cell cycle and these genes are expressed during G1/S/M phases [25]. Cyclins and CDK retarders (CDKRs) are the most important regulators, which control the stimulation of CDKs. The 26S proteasome, through ubiquitin-aided proteolysis, is primarily responsible for controlling the quantities of these regulators at important times in the cell cycle [26]. Twenty CDKs and twenty-nine cyclins are found in human cells [27]. Contrary to CDK7-11, which transcribes genes, CDK1-4/6/7 strictly controls cell cycle phases and mitotic events. Across the whole duration of the cell cycle, the production of CDKs varies cyclically [28,29].

Cell cycle checkpoints are biological processes that allow the cell cycle to halt. They monitor DNA stability and ensure that the previous stage of the cell cycle has finished before the next stage can commence [30]. In general, CDKs initiate the cell cycle machinery. CDK4/6, in particular, advances the G1 phase. When these enzymes interact with D-cyclins, such as cyclins D1, D2, and D3, they become active and form G1 kinases [31]. The progression of the G1 phase is significantly influenced by cyclin D1 [32]. The triggered G1 kinases and cyclin E interact to phosphorylate the Rb (pRb) protein [33]. At this moment, the pRb suppresses the inhibitory function of the TF E2F. The event enables the E2F proteins to become activated, which promotes the gene transcriptions for the G1/S switchover and replication of DNA, enabling the cell to enter into the S phase [34]. Nearly all kinds of human tumors appear to share the loss of control at the G1/S transition stage [35]. BRCA1 transcripts are highly active and are hyperphosphorylated at the end of G1, following the G1/S checkpoint. They are then temporarily dephosphorylated just before the M phase. Many of the proteins connected to BRCA1 may be important for each cell cycle phase [30]. A study reported that miR-34a pauses the cell cycle in the G1 phase by downregulating CCND1 and CDK6, which are linked to multiple miR-34a substrates, either cumulatively or interactively [35].

In the SW-480 and HCT-116 cell lines, RT-PCR was used to ascertain how urosolic acid affected the transcription of CCNB1 and its associated genes, including CDK1/2, CCND1, CCNA2/B2, CDC20, and CKS2. Interestingly, these clusters of genes had considerably lower levels of mRNA expression, including CCND1, in comparison to the reference group. Urosolic acid and Ro-3306 treatments dramatically reduced their mRNA expression [36]. Downregulation of CCND1 expression has recently been demonstrated by the rise in pncCCND1_B and the fall in DHX9 synthesis in TC-71 and SK-N-MC cell lines when treated with etoposide. In the CCND1 promoter region, these treatments encourage epigenetic modifications and the formation of DNA–RNA duplexes. Simultaneously, Sam68 interacts with HDAC1 and induces deacetylation of the surrounding chromatin. In the DNA repair reaction, Sam68 functions as a unique signaling protein by connecting the fragmentation of chromatin regions with the regulation of gene expression [37].

The cyclin B1 protein is encoded by CCNB1, which controls the G2/M switchover in the cell cycle process. When CCNB1 and CDK1 form a complex, 13S condensin (13SC) is phosphorylated and stimulated. 13SC facilitates certain initial mitotic activities that result in the generation of compressed chromosomes. Additionally, the mitochondrial CCNB1/CDK1 complex phosphorylates complex I moieties, whose enzyme activity is enhanced as a result. In due course, a raise in respiration in mitochondria, O_2_ expenditure, and an increase in ATP synthesis provide powerful energy to cells. Eventually, G2 to M phase switchover takes place, which reduces the entire progress of the cell cycle [36]. There are 22 TFs that can adhere to the promoter sequence of CCNB1, in accordance with the transcriptional regulatory network. Yet, CCNB1, along with other cell cycle regulators, such as BUB1, TTK, and CDC25C, was also associated with the majority of the TFs. The process of mitosis may be affected by their abnormal expression, and a failure in mitotic switchover is a factor in the development and hostility of pituitary adenomas. As the invasiveness intensified, CCNB1 expression grew progressively [38]. Following the synthesis of CCNB1, CDC20, CDC7, BUB1B, and MCM3 have been examined in hepatocellular carcinoma (HCC) samples. They were associated with higher histologic severity coupled with vascular infiltration. The presence of a low-grade cancer or the absence of cancer were anticipated in HCC patients [39]. Another study reported the escalated synthesis of CCNB1′s mRNA and proteins in patients with HCC [40].

BUB1, BUB3, BUBR1, MAD1, and MAD2 form the mitotic checkpoint complex (MCPC). It is fundamental to the spindle assembly checkpoint (SACP). When metaphase first begins, CDC20 interacts with MCPC. Since SACP is an oligomeric protein, it governs microtubule adhesion to each kinetochore throughout mitosis to stop the formation of cells with faulty or mutated genomes. In the vicinity of detached kinetochores, the SACP delays the change to anaphase from metaphase, blocking the stimulation of anaphase-promoting complexes/cyclosomes (APC/C) [41]. At the onset of mitosis, CDC20 aggregates with the APC/C and activates its ubiquitin ligase activity. This complex facilitates ubiquitination, leading to the breakdown of cyclin B and securin [42]. Once securin is broken down, it liberates separase to split the connection of the centromere. If it is active, then it interacts and halts the function of separase. For separase to work effectively, securin is required [43]. In this way, they enhance the initiation of anaphase and promote mitotic evasion [42]. Chromosome stability may be compromised by securin overexpression or deficiency [44].

During mitosis, PHF8 also engages with the CDC20-carrying APC. Interestingly, an innovative, KEN- and D-box-detached LXPKXLF motif on PHF8 is identified as being necessary for interacting with CDC20. It has been reported that APC polyubiquitylation of PHF8 is impaired by variations in the LXPKXLF motif in a range of experiments. The absence of PHF8 results in an extended G2 phase and faulty mitosis, which is due to the reality that APC frequently targets regulators of the cell cycle. PHF8 is needed for the transcriptional activation of essential G2/M genes during the G2 phase. Collectively, these data suggest that PHF8 is controlled by APC^CDC20^ and is necessary in the G2/M switchover [45]. In osteosarcoma cells, over-synthesis of CDC20 decreased the levels of p21 and Bim protein production. By suppressing apoptosis, it facilitated cellular proliferation, expansion, and infiltration of osteosarcoma cells [46]. Upregulation of CDC20 resulted in HCC and the same was observed in the HCC-associated cell line [39,47,48].

As an SACP, MAD2L1 ensures that the chromosomes are appropriately oriented toward the metaphase plate during cell division [49]. Lung adenocarcinoma patients have a higher risk of cancer relapse and shorter recovery times, due to the elevated expression of MAD2L1 and CDK1 [50]. It has also been demonstrated that MAD2L1 binds with CDC20 and BUB1B and contributes to the abnormal growth of salivary duct carcinoma, despite its participation in signaling transductions [49].

A DNA/RNA cold-shock motif is bound by the Y-box binding protein (YBX1). The motif has been evolutionarily conserved. It regulates mRNA and protein production, as well as DNA repair. It controls various cytoplasmic and nuclear functions. Nuclear YBX1 is essential for the control of transcription through the inverted CCAAT sequence found in the Y-box binding region, whereas cytoplasmic YBX1 affects mRNA solidity and translation. The host’s immune responses against harmful environmental cues, as well as tumor cell growth, survival, and resistance to drugs, are significantly influenced by YBX1. A study demonstrated that enhanced CDC20 expression in breast cancer patients was positively correlated with expression levels of YBX1 [51].

Type I (α and β) and type II (γ) IFNs are a class of multimodal secreted proteins that participate in the modulation of cell development, as well as in antiviral and immunological responses. STAT1 is a key modulator of both types of IFNs. Distinct Janus kinase (JAK) components can be recruited by receptor chains of type I/II IFNs to stimulate both universal and unique STAT proteins. Through specific promoter-responsive regions, they further initiate the formation of a group of IFN-inducible genes. Type I IFNs bind to JAK1 and TYK2 and stimulate STAT1 and STAT2, which leads to the generation of ISGF3. The latter molecule is a unique transcriptional unit that contains p48/IRF-9. In contrast, IFN-γ predominantly binds to JAK1/2 and causes sustained STAT1 stimulation. The activated STAT1 increases gene expression by interacting with gamma-activated sequences (GAS). Additionally, both IFN types stimulate STAT3, perhaps to a lesser degree and for a shorter period [52]. In particular, IFNγ, IL-6, erythropoietin, and growth factors from fibroblasts and the epidermis activate STAT3. They phosphorylate STAT3 at its tyrosine residue located at position 705 (Y705), whether they are mediated by receptors or non-receptor-mediated TKs. Due to this stimulation, STAT3 assembles into homo- or heterodimers that cross the nucleus and attach to particular DNA regions. Eventually, they control several biological functions, including cell multiplication, differentiation, and apoptosis [53]. As demonstrated by a prior study, STAT1 and STAT3 play conflicting roles in the development of tumors. By preventing angiogenesis, tumor progression, and metastasis, and inducing apoptosis, STAT1 functions as a tumor suppressor. Alternately, the STAT3 pathway is connected to the development of cancer [52]. In this next section, we will explore whether or not this speculation makes sense by mapping the results from the KEGG table.

When prolactin (PLN) binds to its receptor, the receptor dimerizes and phosphorylates JAK2. Subsequently, STAT1, STAT3, STAT5a, and STAT5b are recruited and phosphorylated. Activated STAT5a/b dimerizes and moves to the nucleus, which attaches to the GAS in the promoter region of certain genes. PLN also promotes PI3K/AKT and MAPK pathways [54]. Increased expression of PLN and its receptor has been linked to laryngeal, hepatocellular, colorectum, prostate, breast, ovary, and endometrium cancers [55]. Our KEGG results demonstrated the link between STAT1/3 DEGs and the PLN signaling pathway. The results predict that psoriatic people have a possibility of developing any of these cancers in conjunction with gender-based cancer.

The aberrant remodeling of blood vessels brought on by AGEs and their RAGE receptors (AGEs/RAGE axis) leads to vascular complications in patients with diabetes [56]. Cell line and animal models have both been used to study type 2 diabetes. When the RAGE production was increased, it elevated the expression of pJAK2, pSTAT3, pDRP1, and TRPM. Interestingly, the expression of the aforementioned proteins significantly decreased after RAGE inhibition. Hampering the phosphorylation of STAT3 and DRP1 had impacts that were analogous to those caused by RAGE reduction and TRPM expression. The phenotypic shift of vascular smooth muscle cells in diabetic animal models is caused by the RAGE/JAK2/STAT3 pathway in the maintenance of diabetes-mediated circulatory problems through the modification of kinetics in mitochondria [57].

In many malignancies, programmed death-ligand 1 (PD-L1) serves as an immunological barrier. It is well documented that EBV+ gastric cancers express PD-L1 very often. A study reported that IFNγ treatment increases PD-L1 production in EBV+ SNU-719 cells, following the activation of the JAK2/STAT1/IRF-1 axis. IRF-1 adheres to the PD-L1 promoter that contains the IRF-1α region and precisely controls the transcription of PD-L1. Through the overexpression and stimulation of JAK2, the EBV nuclear antigen 1 partly promotes IFNγ-driven PD-L1 expression [58]. The synthesis of LMP-1 was enough to stimulate STAT1 generation, its adherence to DNA, and transcriptional activation in EBV-immortalized cells. The CTAR-1/2 domains in LMP-1 are responsible for the elevation in STAT1 expression [59].

STAT3 is not necessary for the pancreas to develop normally; however, the majority of pancreatic ductal adenocarcinomas (PDA) demonstrate ubiquitous STAT3 expression. Notably, pSTAT3 and gp130 levels were highly associated with one another. The gp130 inhibiting antibodies effectively inhibited tyrosine phosphorylation in STAT3 in the subgroup of PDA cell lines that exhibited robust pathway activation. Human PDAs displayed higher gp130 expression than the normal pancreas. The gp130 protein is a part of the IL-6 receptor complex. Because IL-6 and LIF, two of the gp130 ligands, were also markedly raised in human PDA tissues, it is clear how important this pathway is. Experiments using cell lines showed that STAT3 activity in established PDAs was significantly influenced by autocrine signaling [60]. In addition, pancreatic cancer (PC) growth is aided by irregularly amplified lncRNA PSMB8-AS1 through the enhancement of the STAT1/PD-L1 network. By altering the miR-382-3p/STAT1/PD-L1 axis, PSMB8-AS1 overexpression can encourage STAT1 production and lead to PC malignant tendencies [61].

In ulcerative colitis patients’ mucosal samples, and to a minor extent in those with Crohn’s disease (CD), increased STAT1 expression and activation were reported. In Western blot assays, elevated concentrations of SOCS-3, a suppressor of STAT initiation, were discovered in CD samples and healthy standards, but no variations in SOCS-1 expression were found. In the inflamed mucosa of these patients, neutrophils and monocytic cells were the primary sources of pSTAT1. Administration of systemic glucocorticoids reduced the levels of pSTAT1. Studies conducted in vitro revealed that steroid administration has a direct impact on STAT1 stimulation [62]. STAT3 is momentarily active in healthy cells, but continuously triggered STAT3 is linked to inflammatory bowel disease (IBD), which modifies the response of gut immune cells. One of the IBD vulnerability sites is the STAT3 gene. IBD patients have significant concentrations of IL-6 in their serum and mucosa, and serum IL-6 concentrations can indicate when the condition will reappear. When IBD is active, IL-6 and its dissolvable receptor increase in the lamina propria and cause the activation of the T-cell deficit of IL-6R. This induces STAT3 to switch on Bcl-2 and Bcl-x_L,_ the anti-apoptotic genes. Through increased IL-6 secretion by primed Th1/17 cells, this signaling pathway promotes mucosal T-cell retention and maintains inflammation [63].

By focusing on the miR-33a-5p/KPNA4 pathway, it was observed that STAT3-influenced overexpression of circCCDC66 promoted the development of non-small cell lung cancer (NSCLC) [64]. In NSCLC sufferers, their serum and tumor samples comprised elevated quantities of STAT3-driven cytokines IL-6/11/22 and leptin and HGF growth factors. These molecules support ongoing STAT3 recruitment through auto- and paracrine modes. In addition, the dysregulation of STAT3 controllers, including SOCS, PTP, and PIAS proteins, has been observed in NSCLC tumors and leads to a rise in pSTAT3 quantities [65].

RelA (p65), RelB, c-Rel, NF-κB1 (p50), and NF-κB2 (p52) are the proteins that make up the TF family of nuclear factor-kappa B (NF-κB). To generate transcriptionally functional homo- and heterodimer units, NF-κB molecules attach to one another. The NF-κB transcription complex is coupled to IκB, which ensures that NF-κB is inhibited and confined to the cytoplasm in an inactive form. When inhibitory B kinases phosphorylate IκB, it releases NF-κB, in addition to cytokines, hormones, growth factors, and various triggering signals. In the nucleus, the NF-κB dimers adhere to signature spots and trigger the essential gene transcription entangled in the critical functions of the cells [66]. RelA and c-Rel drive TCR signaling and activate naive T-cells. Notably, they are key to inducing RORγt, a Th17 lineage TF, and help to generate Th17 cells [67].

Immature or quiescent T-lymphocytes are kept in a dormant state by large concentrations of nuclear FoxO1 and FoxO3a, which boost the synthesis of p27^Kip1^. The latter molecule is a CDK inhibitor that supports cell cycle arrest. By increasing IκB production, enhancing IκB-aided localization of NF-κB in the cytoplasm, and blocking IL-2 transcription, FoxO molecules also stop T-cell stimulation and expansion [68]. Nevertheless, during TCR activation, PI3K/AKT phosphorylates and halts FoxO. The scaffold protein 14-3-3 expels FoxO from the nucleus by forming a complex with it and also halts its nuclear entry by disrupting its nuclear localization signal (NLS) [69]. The inhibition of FoxO decreases the levels of p27^Kip1^ and IκB, which leads to the progression of the cell cycle, IL-2 expression, and expansion of T-cells [70]. Indeed, unchecked T-cell stimulation should be prevented. Effector T-cells have been shown to inhibit T-cell proliferation by secreting the anti-inflammatory cytokine IL-10 [71]. Due to IL-6/10 cytokine signaling, the elevated quantities of p/U-STAT3 bind to the N-end of the FoxO1-14-3-3 complex, leading to the generation of a further robust STAT3-pFoxO complex. The pFoxO in the complex attaches to the NLS of STAT3 proteins, and thus travels back to the nucleus. By doing so, STAT3 acts as a doorkeeper for T-cell activation [70].

The neurological condition labeled HTLV-I-associated myelopathy and adult T-cell leukemia/lymphoma (ATCL) are both caused by the human T lymphotropic virus type I (HTLV-I). As a CD4+ T-cell malignancy, ATCL cells frequently exhibit complicated aneuploidy, including trisomy 3/7, sectional excision of 6q, and anomalies of 14q11, in contrast to cells from all other leukemias [72,73]. Although it is yet to be properly explained, Tax, the oncoprotein from HTLV-1, has been related to the progression from invasion to tumorigenesis. Tax induces NF-κB initiation, cell cycle disruption, and cell modification, while mediating the induction of viral transcription and changing the host cell’s processes in a pleiotropic manner [73]. Tax culminates in mitotic abnormalities that are accompanied by an early and abrupt decline in cyclin B1 and securin quantities, which is facilitated by APC/C^CDC20^. This is in line with the notion that Tax stimulates abnormal APC/C^CDC20^ recruitment to prevent the obstruction of mitotic escape and advancement of aneuploid cells, which are prevalent in ATCL [74]. Although ATCL is a type of lymphoid neoplasm, HTLV-1 also mediates myeloid neoplasms and hematological malignancies [41].

To control the transcription of those genes implicated in cell development and proliferation, MYC generates the TF Myc, which forms a dimer with Max and attaches itself to the 5′-CANNTG-3′ sequence present on the E-boxes. Adenomatous polyposis coli (ADPOC) and the WNT pathway work together to control β-catenin, which, after nuclear import, takes part in transactivating MYC. As a consequence, when ADPOC is lost, MYC expresses itself aberrantly. Severe oncogenic MYC expression activates p53 or Arf when MYC is dysregulated by the loss of upstream regulators, such as APC, gene duplication, or chromosomal translocation. For instance, p53 or Arf mutations that lead to checkpoint regulatory loss reveal MYC’s oncogenic power [75].

TP53 is a tumor suppressor gene, whose p53 protein consists of a DNA binding region at its core, flanked by a transcriptional stimulation region at the N-end and a regulatory, tetrameric region at the C-end. MDM2, a potent inhibitor, usually attaches to p53 and tends to result in protein degradation. Protein kinases are upregulated in response to radiation-induced stress and oncogene-mediated DNA damage, which hinders MDM2. p53 that has been released is stabilized and activated. Cell cycle arrest caused by their buildup activates DNA repair genes. The synthesis of BAX and CDKN1A causes cellular senescence and apoptosis if DNA repair is ineffective. When TP53 is defective or lost, cell growth continues amid DNA disruption, which promotes the formation of further mutations. DNA binding adds a further degree of control. The DNA-binding region is typically inhibited by the C-end region; however, acetylation or phosphorylation of the C-end residues can increase DNA binding [76].

A combination of genes and their products constitutes the p53 pathway, which is designed to react to a range of internal and external signals. Homeostatic systems monitor DNA replication, segregation of chromosomes, and mitotic division. They are impacted by these stress signals [77]. Since p53 repairs DNA, halts the cell cycle, regulates senescence, and apoptosis, it is crucial for maintaining genomic solidity and preventing tumors [78]. In HCC, upregulation of the p53 signaling pathway proteins CCNB1, CDC20, and CENPF has been frequently observed. The expression rates of Th1/17 cytokines, namely IFN-γ, IL-17, and TNF-α, in peripheral blood were strongly associated with these genes. Meanwhile, a clear association between the synthesis of CENPF and the amount of CD8+ T-cells in peripheral blood and an opposing association between the synthesis of CENPF and the amount of CD4+ T-cells were reported. The transcript quantities of PD-1, TIM-3, and CTLA-4 were reported among the repressive checkpoint molecules whose relationships with those three genes in the HCC microenvironment were favorable [79].

The underlying agent of Kaposi’s sarcoma (KAS), the most recurrent cancer among people with AIDS globally, is the KAS-associated herpesvirus (KASAHV) or human herpesvirus 8. KASAHV is linked to primary effusion lymphoma (PELA), and multifocal Castleman’s disease (MCAD). PELA is a B-cell lymphoma that occurs in the pleural cavity, whereas MCAD is a B-cell illness in the lymph nodes. KASAHV is found in spindle cells, the dominant cancer cell group, in KAS malignancies. Spindle cells have lymphatic endothelium-specific characteristics and are endothelial in nature. Almost all spindle cells sustain latent infection of KASAHV, even though only a small number of cells constantly suffer from lytic onset [80]. In the KASAHV lytic cycle, ORF50, a counterpart of the RTA gene product of the Epstein–Barr virus, triggers immediate and delayed gene transcription. The expression of ORF50 is mainly connected with KASAHV-related pathologies. Using a CBP-induced approach, it can be observed that ORF50 inhibits p53-related apoptosis by interacting with CBP and histone deacetylase in the cell. IL-6, v-Src, and reporter genes of STAT activate ORF50, which attaches to STAT3 by transactivating STAT3′s C-end domain and numerous other STAT3 regions. By evading the phosphorylation of tyrosine residues, it recruits STAT3 and causes its monomers to form dimers inside the nucleus [81].

Acute myeloid leukemia (AMLA) is a very diverse disease and has an unpredictable prognosis. It is the cause of around 80% of the overall issues associated with leukemia and represents the most common type of adult leukemia. Bone marrow failure and unsuccessful erythropoiesis are caused by the duplicate growth of premature blast cells in the bone marrow and circulating blood. Chromosomal translocations, genetic defects, or alterations at the molecular level can all induce AMLA, whose pathogenesis is closely associated with mutations in the NPM1, FLT3, RUNX1, IDH, and TP53 genes [82]. In human AMLA, STAT3 is necessary for transcriptional control. The root causes of the increased continuous STAT3 expression in AMLA cells seem to differ amongst susceptible people. Constitutive signaling via an upstream route is one cause, in addition to the occurrence of mutations following STAT3 activation. The amino acid residues from 585 to 688 in the SH2 domain are required to form dimers and can enhance the transcription of STAT3. Mutations have been reported in these domains, especially in patients with lymphocytic leukemia, as well as hepatocellular adenomas. Both of these malignancies demonstrated mutations in Y640F. In-frame insertions in codons 657–658 were observed in patients with adenomas, along with mutational hotspots at D502Y, D661V, D661Y, E166Q, K658Y, L78R, and N647I [83].

In addition to STAT3, STAT1 can also trigger tumorigenesis in psoriasis patients, together with other genes that participate in the cell cycle events.

The existing biologics either reduce the intensity of psoriasis or malignancies based on their stage. The current studies are limited to the role of biologics in reducing malignancies, as well as psoriasis in psoriasis patients. However, various studies have reported the initiation or progression of distinct cancers in these patients. The mechanisms associated with the signaling pathways impacted by biologics that can assist in treating cancer and psoriasis need attention, even though biologics have been used to address both conditions. However, this study highlights acitretin, one of the biologics annotated here as being effective against STAT3. Its mode of action, which can decrease the myeloid form of cancer in the psoriasis model, is summarized (Table 11) [84].

## 5. Conclusions

As demonstrated in the performed study, different forms of cancers have been linked to the dysregulation of cell cycle genes and STAT1/3. These hub genes are regulated by seven TFs, including STAT3. Among them, it is possible to target BRCA1, since it regulates the majority of the hub genes, followed by RELA and TP53. Targeting remedies for hub genes on the one hand and identifying remedies for TFs on the other hand can help us to treat the comorbidities in psoriasis sufferers more efficiently. STAT3 itself acts as a hub gene, as well as a TF. Since it also triggers different cancers due to its upregulation, one should also focus on controlling its expression.

Bountiful biologics exist that can be used to treat psoriasis and its associated comorbidities. However, treating both the associated comorbidity and cancer in parallel with psoriasis is a challenging scenario. To address the proper regulation of the hub genes mentioned above, a total of 147 different remedies against these genes and their TFs were screened. They must, however, be tested in psoriatic cell lines and animal models that show particular comorbidities. Numerous trials must be conducted to effectively treat psoriasis patients without exacerbating the comorbidity. The therapeutic interventions must be tailored so that both the dermal disorder and comorbidity can be treated. This situation demands additional and expandable research to understand which remedy will reduce both conditions, without causing any further side effects in psoriasis patients.

## Figures and Tables

**Figure 1 cells-11-03867-f001:**
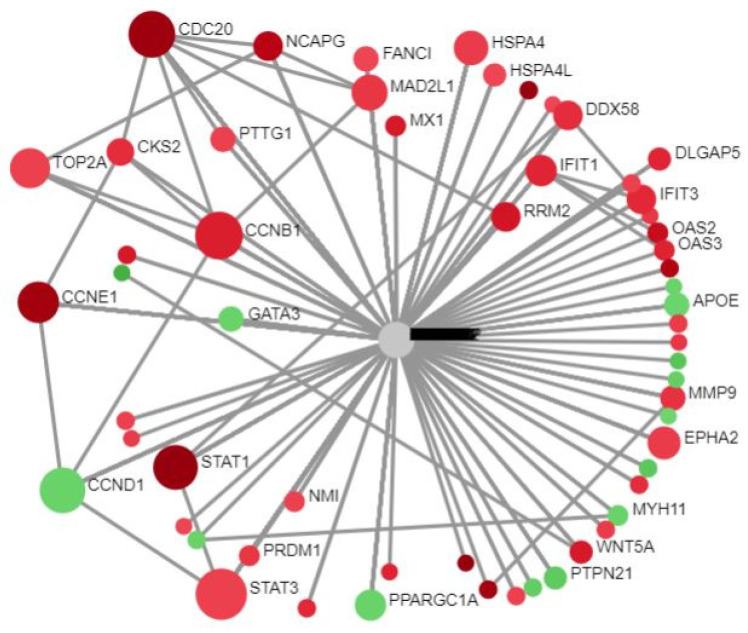
Circular bipartite view of SN1.

**Figure 2 cells-11-03867-f002:**
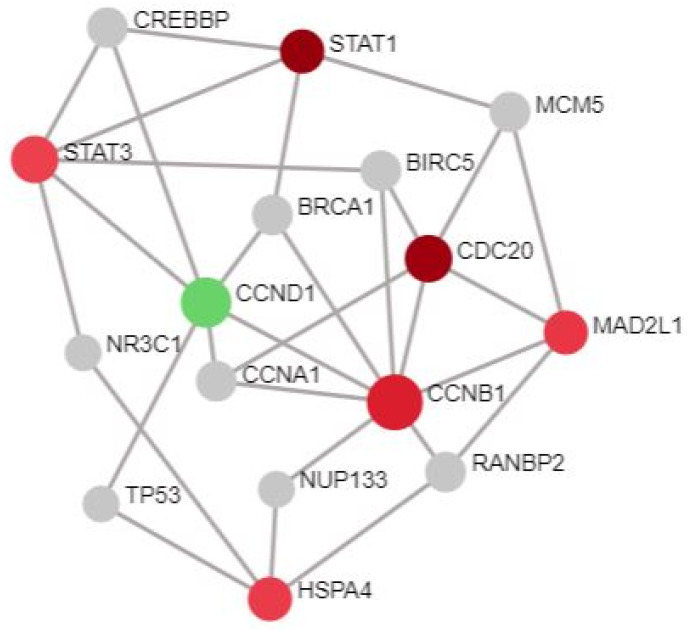
Hub genes from SN1.

**Table 1 cells-11-03867-t001:** Annotation of hub genes in GO BP.

GO BP Terms	Hits	URGs	DRGs	*p* Value	FDR
Mitotic cell cycle checkpoint	4	CCNB1, MAD2L1, CDC20	CCND1	3.88 × 10^−7^	0.000318
Cell cycle checkpoint	4	CCNB1, MAD2L1, CDC20	CCND1	4.89 × 10^−6^	0.00201
Regulation of mitotic cell cycle	4	CCNB1, MAD2L1, CDC20	CCND1	1.18 × 10^−5^	0.00298
Negative regulation of cellular component organization	4	CCNB1, MAD2L1, STAT1, CDC20	Nil	1.46 × 10^−5^	0.00298
Cell cycle arrest	4	CCNB1, MAD2L1, CDC20	CCND1	2.59 × 10^−5^	0.00424
Cell division	4	CCNB1, MAD2L1, CDC20	CCND1	5.04 × 10^−5^	0.00651
Negative regulation of cell cycle	4	CCNB1, MAD2L1, CDC20	CCND1	5.56 × 10^−5^	0.00651
Positive regulation of cell proliferation	4	CCNB1, STAT1, CDC20	CCND1	0.000278	0.0265
Mitotic sister chromatid segregation	2	CCNB1, MAD2L1	Nil	0.000291	0.0265
Sister chromatid segregation	2	CCNB1, MAD2L1	Nil	0.000324	0.0266
Negative regulation of cellular metabolic process	5	CCNB1, STAT3, MAD2L1, STAT1, CDC20	Nil	0.000365	0.0268
Regulation of cell cycle	4	CCNB1, MAD2L1, CDC20	CCND1	0.000442	0.0268
Response to drug	3	CCNB1, STAT3	CCND1	0.000451	0.0268
Reproduction process	5	CCNB1, STAT3, STAT1, CDC20	CCND1	0.000457	0.0268
Negative regulation of metabolic process	5	CCNB1, STAT3, MAD2L1, STAT1, CDC20	Nil	0.000563	0.0297
Reproduction	5	CCNB1, STAT3, STAT1, CDC20	CCND1	0.000614	0.0297
Mitotic cell cycle	4	CCNB1, MAD2L1, CDC20	CCND1	0.000621	0.0297
Regulation of developmental process	5	CCNB1, STAT3, STAT1, CDC20	CCND1	0.000651	0.0297
Cell proliferation	5	CCNB1, STAT3, STAT1, CDC20	CCND1	0.000693	0.0299
Regulation of cyclin-dependent protein kinase activity	2	CCNB1	CCND1	0.00079	0.0316
Mitosis	3	CCNB1, MAD2L1, CDC20	Nil	0.000808	0.0316
Cell cycle phase	4	CCNB1, MAD2L1, CDC20	CCND1	0.000921	0.0333
M phase of mitotic cell cycle	3	CCNB1, MAD2L1, CDC20	Nil	0.000969	0.0333
Regulation of mitosis	2	MAD2L1, CDC20	Nil	0.000976	0.0333
Negative regulation of cellular protein metabolic process	3	CCNB1, MAD2L1, CDC20	Nil	0.00107	0.0352
Positive regulation of cell cycle	2	CCNB1	CCND1	0.00127	0.04
JAK-STAT cascade	2	STAT3, STAT1	Nil	0.00145	0.0435
Cellular protein catabolic process	3	CCNB1, MAD2L1, CDC20	Nil	0.00149	0.0435
Regulation of protein modification process	4	CCNB1, MAD2L1, CDC20	CCND1	0.00163	0.0444
Negative regulation of cell differentiation	3	STAT3, STAT1	CCND1	0.00168	0.0444
Negative regulation of protein metabolic process	3	CCNB1, MAD2L1, CDC20	Nil	0.00168	0.0444

**Table 2 cells-11-03867-t002:** Annotation of hub genes in GO MF.

GO MF Terms	Hits	URGs	DRGs	*p* Value	FDR
Enzyme binding	5	CCNB1, STAT3, STAT1, CDC20	CCND1	7.03 × 10^−5^	0.0273

**Table 3 cells-11-03867-t003:** Annotation of hub genes in GO CC.

GO CC Terms	Hits	URGs	DRGs	*p* Value	FDR
Spindle pole	3	CCNB1, MAD2L1, CDC20	Nil	7.42 × 10^−6^	0.00167
Cytosol	6	CCNB1, STAT3, MAD2L1, STAT1, CDC20	CCND1	1.00 × 10^−4^	0.00951
Spindle	3	CCNB1, MAD2L1, CDC20	Nil	0.000127	0.00951
Nucleoplasm	5	CCNB1, STAT3, STAT1, CDC20	CCND1	0.000271	0.0152
Nuclear part	6	CCNB1, STAT3, MAD2L1, STAT1, CDC20	CCND1	0.000369	0.0166

**Table 4 cells-11-03867-t004:** Annotation of hub genes in KEGG.

KEGG Pathways	Hits	URGs	DRGs	*p* Value	FDR
Cell cycle	4	CCNB1, MAD2L1, CDC20	CCND1	2.12 × 10^−6^	0.000674
Prolactin signaling pathway	3	STAT3, STAT1	CCND1	2.42 × 10^−5^	0.00316
Pancreatic cancer	3	STAT3, STAT1	CCND1	2.98 × 10^−5^	0.00316
AGE-RAGE signaling pathway in diabetic complications	3	STAT3, STAT1	CCND1	7.06 × 10^−5^	0.00561
Oocyte meiosis	3	CCNB1, MAD2L1, CDC20	Nil	0.000137	0.00838
FoxO signaling pathway	3	CCNB1, STAT3	CCND1	0.000162	0.00838
Measles	3	STAT3, STAT1	CCND1	0.000184	0.00838
Hepatitis C	3	STAT3, STAT1	CCND1	0.00026	0.0103
Jak-STAT signaling pathway	3	STAT3, STAT1	CCND1	0.000296	0.0105
Kaposi’s sarcoma-associated herpesvirus infection	3	STAT3, STAT1	CCND1	0.000446	0.0142
Epstein–Barr virus infection	3	STAT3, STAT1	CCND1	0.00056	0.0148
Viral carcinogenesis	3	STAT3, CDC20	CCND1	0.00056	0.0148
HTLV-I infection	3	MAD2L1, CDC20	CCND1	0.00072	0.0176
Inflammatory bowel disease (IBD)	2	STAT3, STAT1	Nil	0.00142	0.0291
Acute myeloid leukemia	2	STAT3	CCND1	0.00146	0.0291
Non-small cell lung cancer	2	STAT3	CCND1	0.00146	0.0291
p53 signaling pathway	2	CCNB1	CCND1	0.00174	0.0326

**Table 5 cells-11-03867-t005:** Annotation of hub genes in Reactome.

Reactome Pathways	Hits	URGs	DRGs	*p* Value	FDR
Phosphorylation of Emi1	2	CCNB1, CDC20	Nil	1.01 × 10^−5^	0.0104
APC/C:Cdc20-mediated degradation of mitotic proteins	3	CCNB1, MAD2L1, CDC20	Nil	2.77 × 10^−5^	0.0104
Activation of APC/C and APC/C:Cdc20-mediated degradation of mitotic proteins	3	CCNB1, MAD2L1, CDC20	Nil	2.88 × 10^−5^	0.0104
Regulation of APC/C activators between G1/S and early anaphase	3	CCNB1, MAD2L1, CDC20	Nil	3.74 × 10^−5^	0.0104
APC/C-mediated degradation of cell cycle proteins	3	CCNB1, MAD2L1, CDC20	Nil	4.45 × 10^−5^	0.0104
Regulation of mitotic cell cycle	3	CCNB1, MAD2L1, CDC20	Nil	4.45 × 10^−5^	0.0104
Interleukin−6 signaling	2	STAT3, STAT1	Nil	6.10 × 10^−5^	0.0122
Resolution of sister chromatid cohesion	3	CCNB1, MAD2L1, CDC20	Nil	0.000104	0.018
Signaling by FGFR1 fusion mutants	2	STAT3, STAT1	Nil	0.000127	0.018
Mitotic prometaphase	3	CCNB1, MAD2L1, CDC20	Nil	0.000129	0.018
Cell cycle checkpoints	3	CCNB1, MAD2L1, CDC20	Nil	0.000141	0.018
Growth hormone receptor signaling	2	STAT3, STAT1	Nil	0.000169	0.0197
Cell cycle (mitotic)	4	CCNB1, MAD2L1, CDC20	CCND1	0.000192	0.0202
Inactivation of APC/C via direct inhibition of the APC/C complex	2	MAD2L1, CDC20	Nil	0.000217	0.0202
Inhibition of the proteolytic activity of APC/C required for the onset of anaphase by mitotic spindle checkpoint components	2	MAD2L1, CDC20	Nil	0.000217	0.0202
Mitotic spindle checkpoint	2	MAD2L1, CDC20	Nil	0.000234	0.0205
APC/C:Cdc20-mediated degradation of cyclin B	2	CCNB1, CDC20	Nil	0.00027	0.0223
APC-Cdc20-mediated degradation of Nek2A	2	MAD2L1, CDC20	Nil	0.000309	0.0228
Signaling by FGFR1 mutants	2	STAT3, STAT1	Nil	0.000309	0.0228
Cell cycle	4	CCNB1, MAD2L1, CDC20	CCND1	0.000439	0.0308
Signaling by FGFR mutants	2	STAT3, STAT1	Nil	0.000655	0.0437

**Table 6 cells-11-03867-t006:** Drugs that target hub genes.

DEGs	Remedies	Roots	PubMed Identifiers
STAT3	Acitretin	1	-
	Pyrimethamine	2	25984755
	Digitoxin	2	-
	Niclosamide	2	-
	Digoxin	2	-
	Ouabain	2	-
CCND1	Nifedipine	3	10051745
	Palbociclib (inhibitor)	2, 4	24417566
	Progesterone	3	16123159
	Lapatinib	5	-
	Cetuximab	5	22117530, 16788380, 18349392
	Methotrexate	3, 5	12972956, 16870553
	Bortezomib	6	20578819
	Abemaciclib	7	
	Acetaminophen	3	11896290
	Tamoxifen	3, 6	12469160, 12602925, 15138475
	Fluorouracil	5	23567490
	Ribociclib	7, 6	29306020

1—TTD, 2—DTC, 3—NCI, 4—Chembl Interactions, 5—PharmGKB, 6—CIViC; 7—Clearity Foundation Biomarkers.

**Table 7 cells-11-03867-t007:** TFs that govern the hub genes.

Name of TFs	TF’s ID in TRRUST	Hub Gene Hits
BRCA1	672	CCND1, CCNB1, STAT3, MAD2L1, STAT1
RELA	5970	CCND1, CCNB1, STAT3, STAT1
TP53	7157	CCND1, CCNB1, STAT3, STAT1
MYC	4609	CCND1, CCNB1, HSPA4
STAT3	6774	CCND1, HSPA4, STAT1
PHF8, YBX1	23979597, 20596676 (PMID)	CDC20

**Table 8 cells-11-03867-t008:** Drugs that target TFs.

TF Gene	Drugs that Target TFs	Roots	PubMed IDentifiers
BRCA1	Everolimus	8	26546619
	Doxorubicin hydrochloride	2	-
	Carboplatin	7, 8, 6	25847936, 25824335, 21135055, 27998224
	Denosumab	8	27322743
	Rucaparib	7, 8, 6, 5, 9	28588062, 27908594, 26779812, 27002934, 27454289
	Acriflavine	2	-
	Thiabendazole	2	-
	Dipyridamole	2	-
	Cisplatin	7, 8, 3, 6	29338080, 25847936, 25193512, 26801247, 16982732, 25072261, 27454289
	Cyclophosphamide	8	25589624
	Temozolomide	8	-
	Irinotecan	8	26842236
	Chlorambucil	8	25193512
	Oxaliplatin	7, 6	25072261
	Niraparib	7, 8, 5, 9	27717299, 23810788
	Olaparib	7, 8, 6, 5, 9	31157963, 23346317, 26546619, 20609467, 27454287, 25193512, 30345884, 25366685, 19553641, 28792849, 25218906, 28578601, 21862407, 22172724, 30797618, 31538027, 24882434
	Paclitaxel	8, 3	12684687
	Tamoxifen	3, 5	11130383, 15197194, 16331614, 15750629, 16636335
	Gemcitabine	3, 6	29338080, 12684687
	Vinorelbine	8, 3	26801247, 14559807
	Daunorubicin hydrochloride, riboflavin, tiaprofenic Acid	2	-
	Talazoparib	8, 6, 5, 9	26546619, 23881923, 28242752
	Mitoxantrone	3	12684687
	Doxorubicin	3	12698198
	Bleomycin	3	14559807
RELA	Artesunate	2	25074847
	Voriconazole	2	18625774
	Gefitinib	5	31664190
	Dexamethasone	2	23219855
TP53	Ibrutinib	8	26563132
	Furazolidone	2	-
	Chloroxine	2	-
	Sertraline	2	16680159
	Cladribine	2	-
	Panitumumab	8	28514312
	Mechlorethamine hydrochloride	2	-
	Hexachlorophene	2	-
	Maprotiline	2	-
	Granisetron	10	-
	Vemurafenib	8	26343583, 28514312
	Doxorubicin hydrochloride	2	-
	Mitomycin	6	14514923
	Hydralazine hydrochloride	2	-
	Triamterene	2	-
	Progesterone	3	16684279
	Carboplatin	8, 6	25567130, 25658463, 11595686, 26494859, 27998224
	Clemastine	2	-
	Tamoxifen citrate	2	-
	Azacitidine	2	-
	Duvelisib	8	-
	Anisindione	2	-
	Trametinib	8	27659046
	Cetuximab	6	24957073
	Pembrolizumab	8	28039262
	Salmeterol xinafoate	2	-
	Chlorpromazine	2	-
	Nortriptyline	2	-
	Loperamide	2	-
	Epirubicin	8, 5	17388661, 22903472
	Trifluridine	8	25700705
	Methotrexate	2, 6	17363498
	Clomipramine	2	-
	Benzalkonium chloride	2	-
	Sertraline hydrochloride	2	-
	Alpelisib	8	27659046
	Fenofibrate	2	16680159
	Cisplatin	8, 3, 6	25376608, 25567130, 9600935, 11595686, 25964101, 23428903, 26086967, 11812076, 27179933, 8678559, 14514923, 23839309, 28652249, 18618574, 26294215
	Bortezomib (inhibitor)	11, 6	28679691
	Prochlorperazine	2	-
	Cyclophosphamide	8, 5	17388661, 16243804, 26438783
	Azathioprine	2	-
	Abemaciclib	8	27217383
	Triflupromazine	2	-
	Temozolomide	8, 6	21730979, 24248532
	Amoxapine	2	-
	Ribavirin	2	-
	Methylene blue	2	-
	Irinotecan	8	25567130
	Niclosamide	2	-
	Bevacizumab	8	27466356, 23670029, 17145525
	Nitazoxanide	2	-
	Chlorambucil	2	-
	Sirolimus	8, 3	26144316, 16651424
	Oxaliplatin	8, 6	24957073, 21468686
	Cinnarizine	2	16680159
	Ethopropazine hydrochloride	2	-
	Propylthiouracil	3	7790147
	Menadione	2	-
	Triclocarban	2	-
	Olaparib	8	22172724
	Melphalan	2	-
	Mercaptopurine	2	-
	Erlotinib	8	27659046
	Paroxetine hydrochloride	2	-
	Paclitaxel	8, 3, 6	16459017, 24065105
	Dopamine	2	-
	Apomorphine	2	-
	Clofibrate	2	16680159
	Tamoxifen	2, 6	10786679
	Fluorouracil	2, 5	-
	Gemcitabine	8	27167172, 23520471, 21389100, 27815358, 26228206
	Venetoclax	5, 12	-
	Thimerosal	2	-
	Enalapril	3	16900775
	Daunorubicin hydrochloride	2	-
	Warfarin	2	-
	Crizotinib	8	25971938, 27149990, 26438783
	Prochlorperazine edisylate	2	-
	Lorlatinib	8	28285684
	Pazopanib	8, 6	26646755, 25669829
	Dabrafenib	8	27659046
	Econazole nitrate	2	-
	Raloxifene	2	-
	Haloperidol	3	16476148
	Sulconazole nitrate	2	-
	Etoposide	8, 6	25964101, 14514923, 24065105
	Daunorubicin	8	16243804
	Clotrimazole	2	16680159
	Encorafenib	8	-
	Mitoxantrone	2	-
	Mitoxantrone hydrochloride	2	-
	Doxorubicin	2, 8, 6	25658463, 21399868, 16243804, 23165797, 26826118, 26288684, 22698404, 17363498, 9569050, 24065105
	Trifluoperazine	2, 3	12415616
	Cytarabine	8	-
	Pimozide	2	-
	Ifosfamide	8	23165797
	Dasatinib	8	26855149
	Capecitabine	6	24957073
	Methimazole	3	7790147
	Methylprednisolone	3	16684279
	Fluphenazine	2	-
	Docetaxel	8, 6	21399868, 22425996
	Perphenazine	2	-
	Vorinostat	8, 6	26009011, 25669829
	Topotecan	8	26438783
	Rituximab	3	11895917
MYC	Ibrutinib	6	28830912
	Bromocriptine	3	11680511
	Azacitidine	3	9006118
	Cetuximab	3	3085922
	Calcitriol	3	8490200, 15598784
	Amifostine	3	9450496
	Cisplatin	6	28490518
	Indomethacin	3	10403534
	Estrone	3	12520970, 12567859
	Thyrotropin	3	3125035, 3726540
	Sulindac	3	12414619
	Olaparib	6	28490518
	Glutamine	3	16898871
	Melatonin	3	7629697
	Imatinib	3	15517875
	Thrombin	3	3023371
	Verapamil	3	1511424
	Methylprednisolone	3	9169090
	Thioguanine	3	1988936
	Vorinostat	3	15583844
	Quinapril	3	9370386

1—TTD, 2—DTC, 3—NCI, 4—Chembl Interactions, 5—PharmGKB, 6—CIViC, 7—Clearity Foundation Biomarkers, 8—JAX-CKB, 9—OncoKB, 10—Clearity Foundation Clinical Trial, 11—TALC; 12—FDA.

**Table 9 cells-11-03867-t009:** Common remedies against multiple TFs.

BRCA1 and TP53	TP53 and MYC	BRCA1, TP53 and MYC
Carboplatin, chlorambucil, cyclophosphamide, daunorubicin hydrochloride, doxorubicin, doxorubicin hydrochloride, gemcitabine, irinotecan, mitoxantrone, oxaliplatin, paclitaxel, tamoxifen, temozolomide	Azacitidine, cetuximab, ibrutinib, methylprednisolone, vorinostat	Cisplatin, olaparib

**Table 10 cells-11-03867-t010:** Common remedies against hub genes and TFs.

STAT3 and TP53	CCND1, BRCA1 and TP53	CCND1 and TP53
Niclosamide	Tamoxifen	Abemaciclib, bortezomib, cetuximab, fluorouracil, methotrexate, progesterone

**Table 11 cells-11-03867-t011:** Role of acitretin in reducing myeloid-derived suppressor cells (MDSCs) in the psoriatic animal model.

Intensity of Psoriasis	Type of Comorbidity	Remedy	Mode of Action of the Remedy
Psoriasis ↑↑	MDSCs ↑↑↑	Acitretin	- Stimulates ERK1/2 MAPK pathway; glutathione synthase increases and glutathione accumulates; glutathione nullifies ROS rate in MDSCs, and thereby lowers the quantity of these cells. - Supports the MDSCs’ differentiation into CD11c+ MHC-II+ dendritic cells and CD206+ M2 macrophages.

↑↑—increase in disease severity; ↑↑↑—increased numbers of MDSCs.

## Data Availability

The overall logFC values of the DEGs across studies can be found in the study by Suárez-Fariñas et al., 2010 [18] at pone.0010247.s002.pdf (83K) (accessed on 31 August 2022).

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
