# Peer review of "Differentially Expressed Cell Cycle Genes and STAT1/3-Driven Multiple Cancer Entanglement in Psoriasis, Coupled with Other Comorbidities"

_cells, 2022, doi:10.3390/cells11233867_

Round 1
Reviewer 1 Report
The authors have presented an interesting research. The overall scope is well met and the methods are appropriately chosen. The discussion is comprehensive. The results are important as a matter of the topic. However, there are a few significant concerns that can be appropriately addressed to improve the paper. My remarks are the following:
- Abstract: there is lack precise aim of the study, it should be added
- Introduction: in my opinion it could be improved. It seems quite chaotic. First paragraph (basic information about skin) is not necessary in my point of view. Information about pathogenesis of psoriasis should be more detailed referring to the topic of the paper
- Discussion is a little bit too long.
Reviewer 2 Report
I believe that the overall composition of the study is fine.
The study is very valuable because it analyzes STAT3 and STAT1 and other cancer-related cascades in many cell lines.
One disappointing point, however, is the lack of mention of the various immunosuppressive biologics that are now being used in psoriasis patients.
When biologics were first used for psoriasis, there was thought to be an increased risk of lymphoma, as in rheumatoid arthritis patients, but in fact they have been found to be no different from those used in healthy controls.
As mentioned in the text, NF-KB, a cancer-causing protein, is regulated by TNF-α. Inhibition of TNF-α is theoretically expected to increase the risk of carcinogenesis, but in actual psoriasis patients, use of TNF-α inhibitors does not increase the carcinogenesis rate.
Please describe any hypothetical reason why the carcinogenicity is not increased in psoriasis patients using biologics.
Even if this question is not resolved, we believe this paper is worthy of publication.
